# Improvement of Strength and Impact Toughness for Cold-Worked Austenitic Stainless Steels Using a Surface-Cracking Technique

**Kwangyoon Kim [1], Minha Park [1], Jaeho Jang [1], Hyoung Chan Kim [1], Hyoung-Seok Moon [1], Dong-Ha Lim [1], Jong Bae Jeon [1], Se-Hun Kwon [2], Hyunmyung Kim [3] and Byung Jun Kim [1,***

[1] Energy Plant R&D Group, Korea Institute of Industrial Technology, Busan 46938, Korea;
kky0858@kitech.re.kr (K.K.); pmh0812@kitech.re.kr (M.P.); jngjho@kitech.re.kr (J.J.);
chancpu@kitech.re.kr (H.C.K.); hyoungseok.moon@kitech.re.kr (H.-S.M.); dongha4u@kitech.re.kr (D.-H.L.);
jbjeon@kitech.re.kr (J.B.J.)

[2] School of Materials Science and Engineering, Pusan National University, Busan 46241, Korea;
sehun@pusan.ac.kr

[3] Department of Nuclear & Quantum Engineering, Korea Advanced Institute of Science and Technology,
Daejeon 34141, Korea; h46kim@kaist.ac.kr

*   Correspondence: jun7741@kitech.re.kr; Tel.: +82-10-5136-7743

**Abstract:** For cryogenic applications, materials must be cautiously selected because of a drastic degradation in the mechanical properties of materials when they are exposed to very low temperatures. We have developed a new technique using a cold-working and surface-cracking process to overcome such degradation of mechanical properties at low temperatures. This technique intentionally induced surface-cracks in cold-worked austenitic stainless steels and resulted in a significant increase in both strength and fracture at low temperatures. According to the microstructure observations, dissipation of the crack propagation energy with surface-cracks enhanced the impact toughness, showing a ductile fracture mode in even the cryogenic temperature region. In particular, we obtained the high strength and toughness materials by a surface-cracking technique at 5% cold-worked specimen with surface-cracks.

**Keywords:** cold-working process; surface-cracking process; impact toughness; strength; low temperatures; austenitic stainless steels

---

## 1. Introduction

The study of very low temperature environments in terms of cryogenics is one field in which the materials play a major part in desirable performances in severe conditions. Cryogenic technologies have various applications in many different fields such as in the power industry, chemistry, electronics, manufacturing, transportation, and food processing by refrigeration [1–7]. For cryogenic applications, materials must be cautiously selected because of a drastic degradation in mechanical properties of the materials when they are exposed to very low temperatures [8]. Generally, austenitic steels with face centered cubic (FCC) structures such as stainless steels [9], Al alloys [10], Ni alloys [11], and Ti alloys [12] are used as low temperature materials because they have high impact toughness at low temperatures [13–15]. Although these alloys with FCC structure have excellent mechanical properties at low temperatures, brittle fracture may occur in welds and rectangular structures due to the stress concentration [16]. A brittle fracture is usually very dangerous because it occurs abruptly with little or no warning, resulting in serious economic losses and potentially a loss of many lives. Therefore, it is

in high demand to develop new technologies in order to improve the mechanical properties of FCC alloys for use in severe environments such as in the cryogenics field.

In particular, materials with good mechanical properties are required for use in severe environments such as the cryogenics field because material characteristics are significantly degraded and changed at low temperatures. Work hardening, also known as strain hardening or cold-working, is a method to strengthen a metal by plastic deformation [17–19]. In detail, the strengthening by cold-working occurs because of a decrease in the mobility of dislocations during plastic deformation of metals. The strain hardening is caused by an increase in the dislocation density within the austenitic structure [20]. Furthermore, the strengthening effect by phase transformation in austenitic stainless steels where the strained-induced martensitic transformation from austenitic phase shows a substantial strengthening effect [21–25]. It has also been found that the yield and tensile strength of austenite stainless steels is gradually improved by increasing the cold-working level. Therefore, a cold-working process is typically considered to be an important technique for increasing the strength of steels [26,27]. However, negative effects such as the unfavorable material embrittlement can be caused by reductions in ductility and impact toughness after a cold-working process [28].

Generally, impact toughness is reduced with an increase in work hardening and precipitates, whereas it typically shows a positive correlation with increases in ductility [29]. In recent research, the grain size refinement was an important technique for improving the impact toughness for high-strength steels [30–34]. Thermal heat treatments affected enhanced impact toughness by an ultra-fine grained structure [29]. Another approach to enhancing impact toughness is to provide multiple pathways for crack propagation, for example, by adding ultra-fine particles within the fine granular structure. Delamination, or splitting, resulting from anisotropic microstructures such as crystalline grains and secondary phases is well known to improve the toughness of metals at low temperatures [23,28,29]. Toughening mechanisms are for distributing the stress near the crack tips by allowing the delamination fracture of the grains from one another, instead of brittle fractures in bulk materials [35–37]. Most engineering designs require materials with high strength and impact toughness to avoid dangerous failure due to brittle fracture at low temperatures [38]. However, it is difficult to obtain tougher and stronger steels because materials typically show opposing characteristics for ductility and brittleness. Therefore, the achievement of a simultaneous enhancement of strength and toughness is a challenge [38,39].

Higher strength and toughness are key requirements for steels used in various structural applications, such as for aircraft, buildings, and heavy machinery including cryogenics applications, in order to satisfy the increasing demands for reliability, durability, and safety. Toughness and strength do not always have opposite characteristics [40,41]. Increasing the toughness of metals without sacrificing other properties is critical for their economic competitiveness [31,38,42,43]. It is true that for intrinsically ductile materials, such as metals, the improvement in strength usually comes at the expense of toughness [44,45]. However, the present results show the possibility and potential of improving toughness and strength at the same time. In this study, we have developed a new technique using a cold-working and surface-cracking process to increase the strength and toughness at the same time. Our proposed method is called a surface-cracking technique which has two processes with the cold-working and surface-cracking process. Cold-working (CW) is one of the typical methods which imparts higher strength to steel, but decreases in ductility accompanied by toughness degradation are generally unavoidable. However, our work shows that enhanced toughness can be achieved using the surface-cracking technique without sacrificing strength. We applied this technique to austenitic stainless steels including base metal and weld metal.

## 2. Materials and Methods

The chemical compositions of stainless steels used in this study are shown in Table 1. A thick plate of STS304 was welded with a STS308L welding electrode (ESAB SEAH, Gyeongnam, Korea) by the submerged-arc welding (SAW) method. The dimension of the welded plate was in the width of

100 mm, the length of 1000 mm, and the thickness of 25 mm. For the post-weld heat treatment (PWHT), the austenitic stainless steels were processed at 105 °C for 2 h, and subsequently quenched in water. In order to observe the microstructures, the test samples were mechanically polished using a 2000 grit SiC paper and then micro-polished using a 3 μm and 1 μm diamond paste. Then, sample surfaces were etched in a solution containing 40% distilled water, 30% $HNO_3$, and 30% $NO_3$.

Tensile testing was carried out at room temperature using a 5 kN full-automatic INSTRON 4204 tensile machine (INSTRON, Norwood, MA, USA) at a cross head speed of 0.03 mm/s. The dimensions of the miniaturized tensile specimens were 5 mm in gauge length, 1.2 mm in width, and 0.5 mm in thickness.

**Table 1.** The chemical compositions of the specimens used in this study (wt %).

| Alloys | C | Mn | Si | Cr | Ni | Mo | Al | Cu | Fe |
|---|---|---|---|---|---|---|---|---|---|
| Base Metal (STS304) | 0.046 | 1.19 | 0.42 | 18.23 | 8.02 | 0.149 | 0.003 | 0.223 | Bal. |
| Weld Metal (STS308L) | 0.02 | 1.98 | 0.41 | 19.71 | 10.79 | 0.03 | - | 0.13 | Bal. |

Vickers hardness tests were performed at room temperature with a load of 4.95 N with 10 s of dwell time. At least 10 measurements were made to calculate the average micro-hardness values by excluding the minimum value and maximum value. We prepared Charpy V-notch impact specimens (10.0 mm × 10.0 mm × 55.0 mm) without or with surface-cracks to evaluate the improvement of impact toughness by surface-cracks (Figure 1). Charpy V-notch impact tests were performed to measure the absorbed energy at temperatures from 20 to −180 °C using a full-automatic Zwick Charpy impact test machine according (Zwick/Roell, Ulm, Germany) to ASTM E23 [46].

Our proposed method is called a surface-cracking technique which involves two processes including a cold-working and a surface-cracking process. The first step was a typical method for improving the strength of stainless steel by a cold-working process. From the as-received (AS) STS304, a plate of steel with 20 mm thickness was cold worked into different thicknesses of the plates CW05, CW10, and CW30, with reductions of plate thickness by 5, 10 and 30%, respectively. In order to minimize the effect of directionality by cold-working, a cold-working process was performed several times at the same thickness reduction rate in both the longitudinal and transverse directions. The configurations and process conditions of the test specimens used in this study are summarized in Table 2.

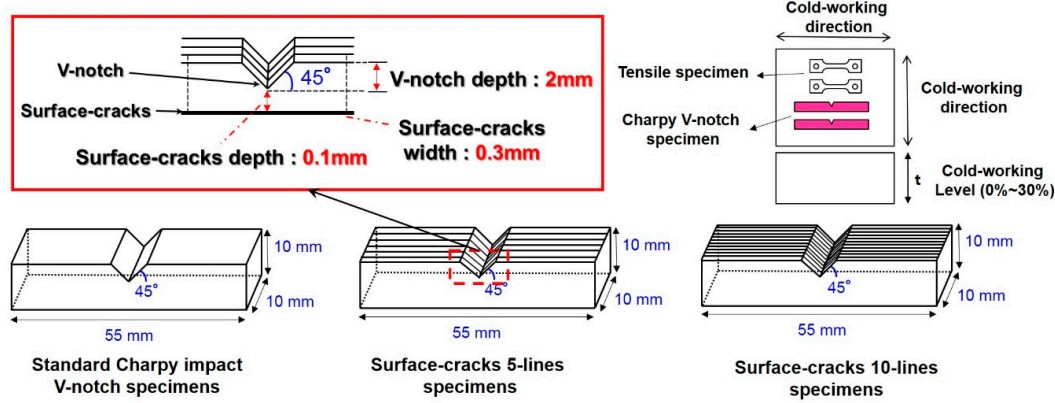

**Figure 1.** Configurations of the specimens used in this study.

The second step was a surface-cracking process for improving the impact toughness of stainless steel by inducing the surface-cracks on the specimen surface. This method was applied to Charpy V-notch impact test specimens to evaluate the enhancement of impact toughness for stainless steels. In this method, surface-cracks were formed in the direction perpendicular to the V-notch direction by wire cut machining V850G (EXCETEK, Taichung, Taiwan). This work was machined to a depth

of 0.1 mm to disperse the tri-axial stress at the V-notch tip using a wire with a width of 0.3 mm (as shown in Figure 1). In order to observe the effect of surface-cracks, surface-cracks were fabricated with 5 and 10 lines and compared with standard specimens without surface-cracks. The overall process of a surface-cracking technique is shown in Figure 2.

**Table 2.** Configurations of the specimens used in this study.

| Specimen ID | Region | Cold-Working Level | Number of Added Surface-Cracks |
|---|---|---|---|
| BM-AS | Base metal | - | - |
| BM-AS-L5 | Base metal | - | 5 lines |
| BM-AS-L10 | Base metal | - | 10 lines |
| BM-CW05 | Base metal | 5% | - |
| BM-CW05-L10 | Base metal | 5% | 10 lines |
| BM-CW10 | Base metal | 10% | - |
| BM-CW10-L10 | Base metal | 10% | 10 lines |
| BM-CW30 | Base metal | 30% | - |
| BM-CW30-L10 | Base metal | 30% | 10 lines |
| WM | Weld metal | - | - |
| WM-L5 | Weld metal | - | 5 lines |
| WM-L10 | Weld metal | - | 10 lines |

To analyze the phase transformation of austenitic stainless steels after a cold-working process, the volume fraction of martensite was measured by a FERITSCOPE FMP30 (Fischer, Windsor, CT, USA). The volume fraction was measured 8 times per specimen in total. The fracture surface of the Charpy impact test specimen was observed with a FE-SEM HITACHI S-4800 (HITACHI, Tokyo, Japan).

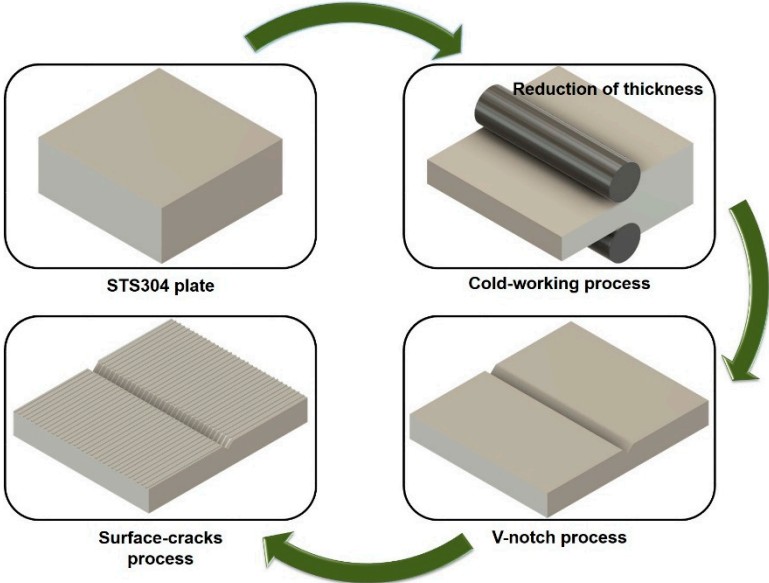

**Figure 2.** Schematic diagram of a surface-cracking technique.

## 3. Results and Discussion

### 3.1. Strengthening by a Cold-Working Process

Figure 3 shows the microstructure of base metal for austenitic stainless steel STS304 before and after a cold-working process. The microstructure of the base metal before cold-working was typically an austenitic structure with 160 μm grain size, as shown in Figure 3a. After the cold-working process, the microstructure was changed from austenite single phase to dual phases which consisted of austenite

and martensite (Figure 3b–d). The fraction of martensitic phase increased with an increase in the cold-working level from 5% to 30%.

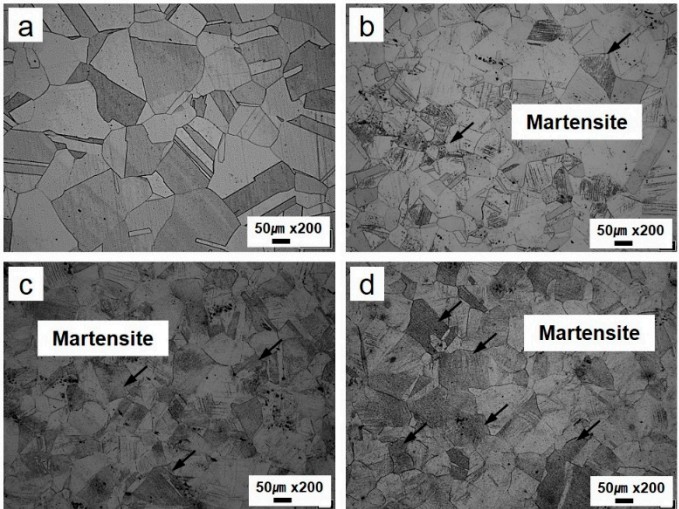

**Figure 3.** The microstructure of the base metal for austenitic stainless steel STS304 before and after a cold-working process: (**a**) AS-BM, (**b**) BM-CW05, (**c**) BM-CW10, and (**d**) BM-CW30.

Figure 4 shows the results of tensile properties for STS304 before and after a cold-working process. The ultimate tensile strength (UTS) of highly cold-worked steel (BM-CW30) with a reduction of plate thickness by 30% was about two times higher than that of the as-received steels (BM-AS). The tensile strength of cold-worked steels increased with an increase of the cold-working level from 5% to 30%. However, the elongation of cold-worked steels decreased with reductions in ductility.

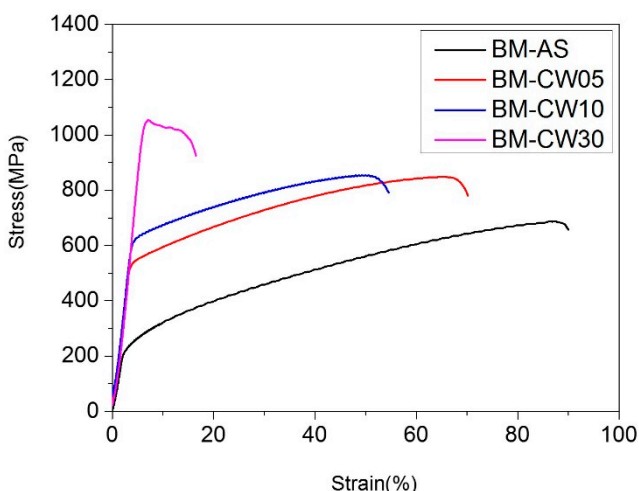

**Figure 4.** The stress-strain curves of STS304 before and after a cold-working process.

### 3.2. Mechanism of Strengthening of a Cold-Working Process

Cold-working generally leads to an increase in the yield strength of stainless steels because of the increased number of dislocations [43]. Figure 5 shows the results of the quantitative analysis for the martensite volume fraction measured by the ferrite scope. As-received steels (BM-AS) with a fully austenite structure were close to a 0% volume fraction of martensite. The volume fraction of martensite increased with the cold-working level from 5% to 30% (Figure 5). In the case of 30% cold-worked specimens (BM-CW30), the volume fraction of martensite increased to about 8%. Therefore, strength

was increased by the phase transformation from austenite to martensite and the increase in dislocations by plastic deformation [20].

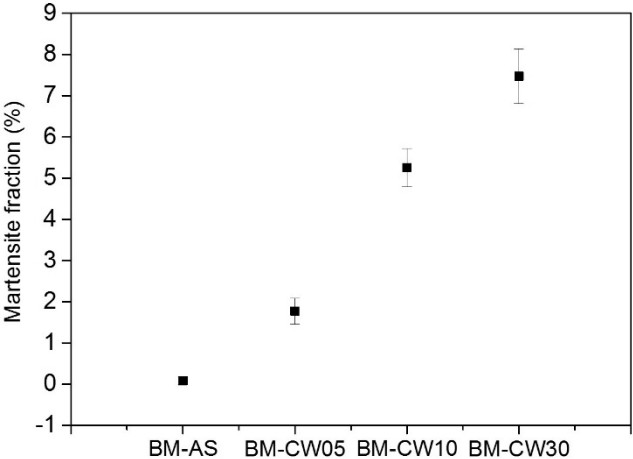

**Figure 5.** Martensite volume fractions (in %) of STS304 before and after a cold-working process.

Figure 6 shows the result of the Vickers hardness test before and after a cold-working process. The hardness value of as-received steel (BM-AS) was about 161 HV. Similar to the tensile test results, hardness values increased with an increase in the level of cold-working. The increase of the hardness value after a cold-working process was owed to the same mechanism as that in the tensile results which have the phase transformation of martensite from austenite and the increase in dislocations by plastic deformation. Therefore, results from tensile and hardness tests were in good agreement with quantitative analysis of the volume fraction of martensite as measured by the ferrite scope. Table 3 summarizes the test results of the tensile properties and Vickers hardness values obtained in this work.

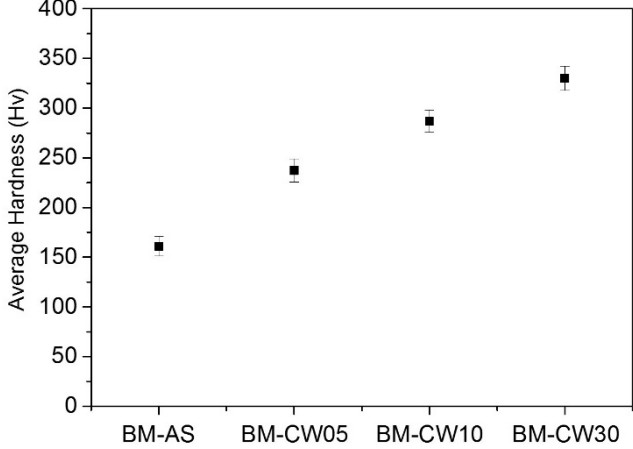

**Figure 6.** Hardness of STS304 before and after a cold-working process.

**Table 3.** Tensile properties and Vickers hardness values of the specimens used in this study.

| Specimen ID | Yield Strength, σy (MPa) | Tensile Strength, σt (MPa) | Elongation, (%) | Average Vickers Hardness (HV) |
|---|---|---|---|---|
| BM-AS | 212 | 689 | 91 | 161 |
| BM-CW05 | 548 | 848 | 70 | 237 |
| BM-CW10 | 611 | 854 | 56 | 286 |
| BM-CW30 | 1002 | 1053 | 17 | 330 |

### 3.3. Toughening by a Surface-Cracking Process

Generally, ordinary cracks in materials weaken their mechanical characteristics due to the stress concentration at crack tips. Although the presence of flaws or cracks causes fractures due to the concentration of stress, a number of aligned cracks in a matrix can distribute the fracture energy in reverse. The absorbed energy vs. test temperature curves for specimens with a surface-cracking process is shown in Figure 7. At room temperature, the absorbed energy of as-received specimens was about 400 J. Although austenite stainless steel with face centered cubic (FCC) structure does not show the ductile-to-brittle temperature (DBTT) behavior, the absorbed energy tends to decrease with decreasing temperature. In the case of specimens with surface-cracks, the absorbed energy was almost the same comparing the as-received specimen at room temperature. However, the specimen with surface-cracks had a higher absorbed energy compared to the as-received specimen without surface-cracks at the low temperature region (below −60 °C) As the number of surface-cracks increased from 0 lines (as-received) to 10 lines, the absorbed energy significantly increased at below −60 °C. At a very low temperature (−180 °C), the absorbed energy of the specimen with 10 surface-cracks (BM-AS-L10) was two times higher than that of the as-received specimen without surface-cracks. Therefore, this effect of enhanced toughness by a surface-cracking process occurred at very low temperatures when the material changed to having brittle properties.

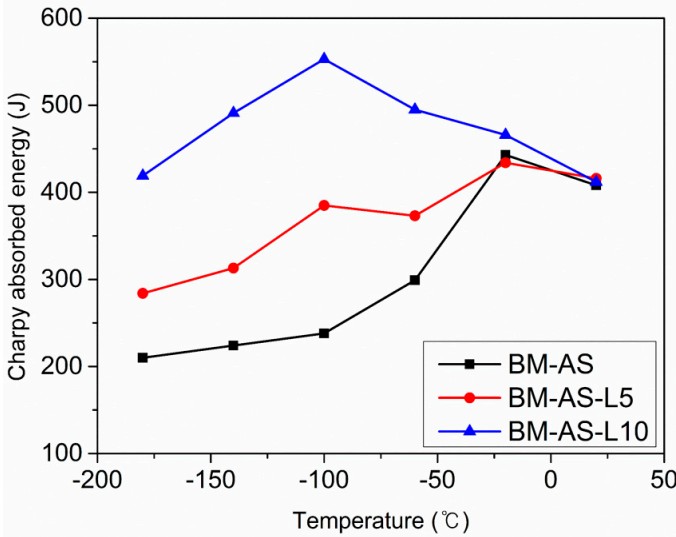

**Figure 7.** The absorbed energy vs. test temperature curves for specimens with surface-cracks.

### 3.4. Toughening by a Surface-Cracking Process

Figure 8 shows the fracture shape of as-received specimens and specimens with surface-cracks after the Charpy impact test. The facture shape of all specimens with or without surface-cracks observed large plastic deformation at room temperature (Figure 8a–c). There was no significant difference between the as-received specimens and specimens with surface-cracks near the V-notch. We also could not observe any differences in plastic deformation by the number of micro-cracks. However, the effect of surface-cracks was clearly observed at the cryogenic temperature (−180 °C) (Figure 8d–f). Compared to the results at room temperature, as-received specimens were completely fractured and accompanied with slight plastic deformation at the cryogenic temperature (Figure 8d). However, the specimens with surface-cracks significantly increased plastic deformation around the V-notch and surface-cracks. The plastic deformation also increased with the increasing number of surface-cracks. From the results of the Charpy impact test in Figure 7, the effect of surface-cracks could be confirmed through the fractured formations. Therefore, the fracture shape of the fractured Charpy specimens was in good agreement with the results of the absorbed energy from the Charpy impact test at room temperature and cryogenic temperature.

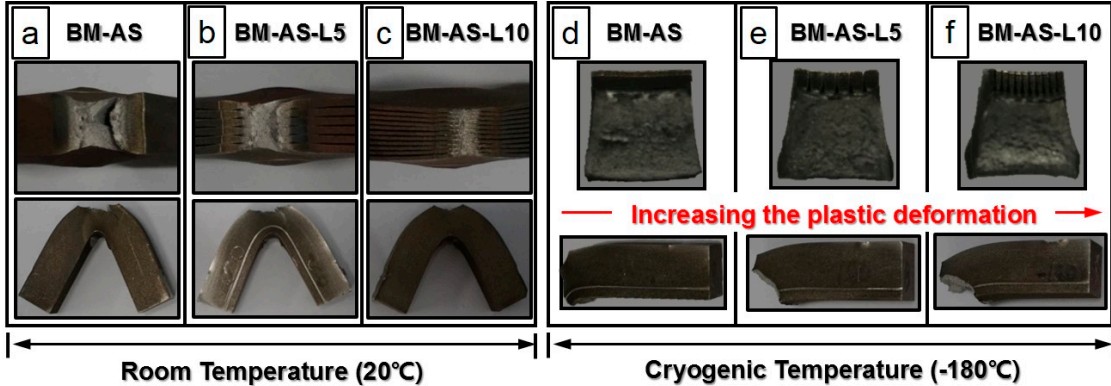

**Figure 8.** The fracture shape of the as-received specimen and the specimen with surface-cracks after Charpy impact test at room temperature (20 °C) and cryogenic temperature (−180 °C): (**a**) BM-AS (20 °C), (**b**) BM-AS-L5 (20 °C), (**c**) BM-AS-L10 (20 °C), (**d**) BM-AS (−180 °C), (**e**) BM-AS-L5 (−180 °C), and (**f**) BM-AS-L10 (−180 °C).

Figure 9 shows the fracture surface of as-received specimens and specimens with surface-cracks at 20 °C. The facture surface of all specimens with or without surface-cracks was typically observed with ductile fracture by large plastic deformation at room temperature. There was no significant difference in the overall fracture surface due to the large deformation around the V-notch in all specimens. The effect of enhanced toughness by a surface-cracking process did not happen in room temperature regions with high ductility properties. Since plastic deformation completely occurred around V-notch tip at room temperature, the additional effect of stress dissipation from the surface-cracks was not significant. Figure 10 shows the fracture surface of as-received specimens and specimens with surface-cracks at −180 °C. The fracture surface of the as-received specimen showed the mixed fracture modes, both ductile and brittle, at −180 °C (Figure 10a) because it became very brittle in the low temperature region despite the ductile behavior at room temperature. However, the fracture mode with surface-cracks was predominantly the ductile mode at very low temperatures (as shown in Figure 10b–c). The rate of ductile fracture increased with the increasing number of surface-cracks. The fracture surface of the as-received specimen observed short stretch zones by blunting at the V-notch tip (Figure 10a, left). However, the fracture surface of specimens with surface-cracks showed the largest extension of stretch zones near the V-notch and surface-cracks, and ductile tearing occurred at very low temperatures (Figure 10c, left). This indicates that vertical surface-cracks in the V-notch direction serve to distribute the stress by increasing the blunting effect at the V-notch tip.

Here, toughening by a surface-cracking process could be explained by the effects of the stress dissipation induced by surface-cracks near the V-notch tip. For example, a number of surface-cracks induced by a surface-cracking process led to the enhancement of impact toughness around the V-notch where the surface-cracks in the vertical direction to the V-notch acted as barriers to propagate the main crack and disperse the principal stresses of the V-notch. As the number of surface-cracks increased, the absorbed energy increased with an increase in the dissipation of the crack propagation energy (Figures 7–10). In other words, impact toughness was enhanced by the change in the stress condition from a plain strain to a plain stress condition due to the reduction of effective thickness by surface-cracks at the V-notch tip. Therefore, the dissipation of the crack propagation energy in the materials with surface-cracks enhanced the impact toughness, showing a ductile fracture mode in the cryogenic temperature region.

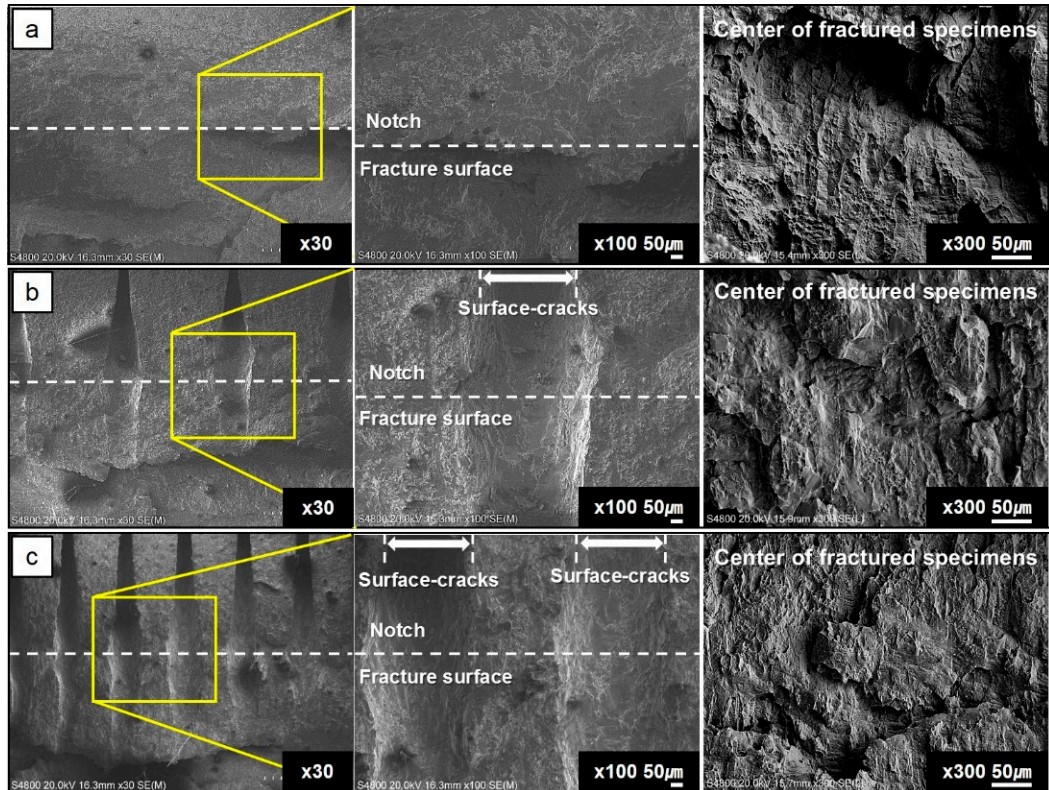

**Figure 9.** The fracture surface of the as-received specimen and the specimen with surface-cracks at room temperature (20 °C): (**a**) BM-AS, (**b**) BM-AS-L5, and (**c**) BM-AS-L10.

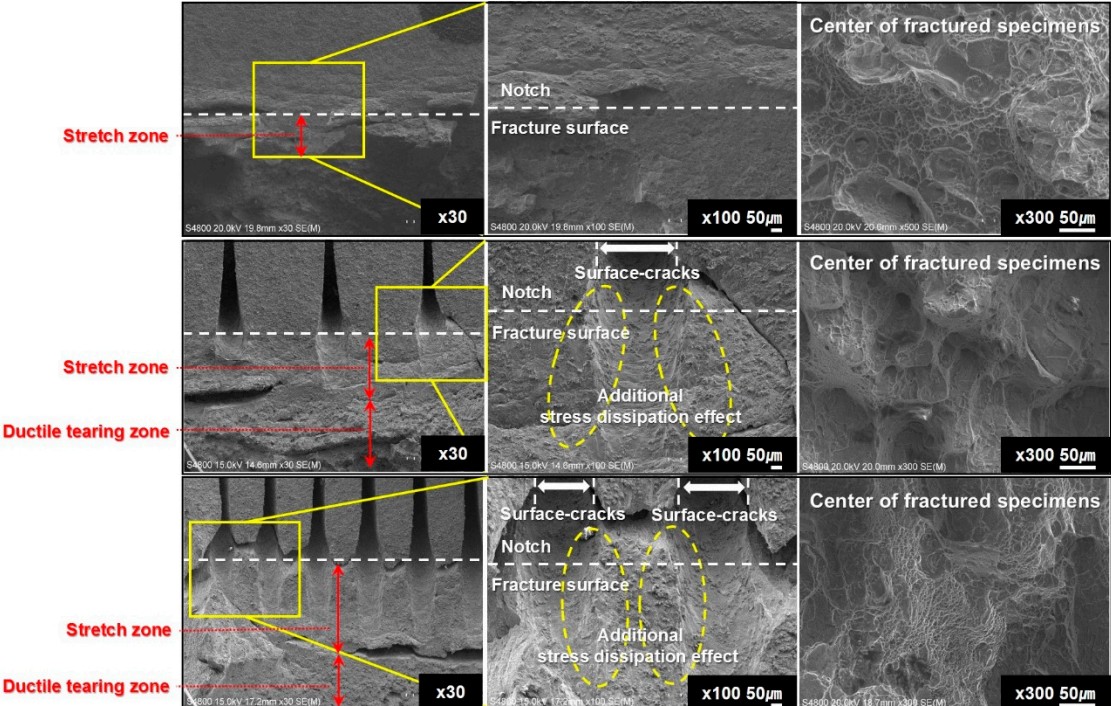

**Figure 10.** The fracture surface of the as-received specimen and the specimen with surface-cracks at cryogenic temperature (−180 °C): (**a**) BM-AS, (**b**) BM-AS-L5, and (**c**) BM-AS-L10.

### 3.5. High Strength and Toughness Steels by a Surface-Cracking Technique

The absorbed energy vs. test temperature curves for cold-worked steels are shown in Figure 11. At room temperature, the absorbed energy of cold-worked steels decreased with a cold-working process. The absorbed energy also decreased with the cold-working level from 5% to 30% (solid lines in Figure 11). As we have already mentioned in the Section 3.2, austenite stainless steels after a cold-working process generally resulted in a higher yield strength and lower ductility owing to the increase in dislocation density and martensitic transformation from the austenite phase. Therefore, the result of the Charpy impact test provided the decrease in absorbed energy due to the reduction of ductility by a cold-working process. When comparing the surface-cracking effect in the same cold-working level, the absorbed energy of the specimen with surface-cracks was about two times higher than that of the specimen without surface-cracks at low temperatures (below −60 °C). In particular, strength and toughness were simultaneously increased in the 5% cold-worked specimen with surface-cracks. The interesting thing was that the effect of enhanced toughness by a surface-cracking technique happened at all temperature regions by excluding the as-received results (BM-AS).

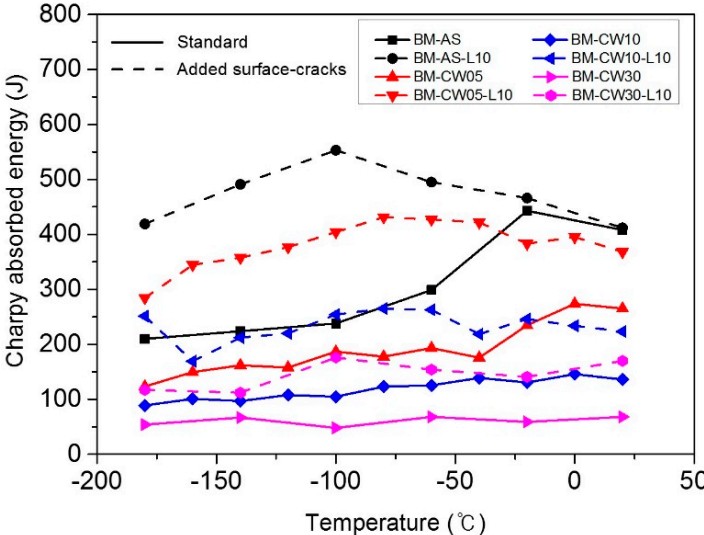

**Figure 11.** The absorbed energy vs. test temperature curves for cold-worked steels with a surface-cracking technique.

Figure 12 shows the fracture surface of cold-worked specimens with surface-cracks at 20 °C. The as-received specimen was fractured in ductile modes by large plastic deformation (Figure 9), while cold-worked steel appeared to be in a mixture of brittle and ductile modes, showing both cleavage facets and dimples (Figure 12a,c,e). The fraction of brittle fracture increased with the cold-working level from 5% to 30% at room temperature. However, in case of the cold-worked steels with surface-cracks, the fraction of the ductile fracture increased due to the effect of surface-cracks after the surface-cracking technique (Figure 12b,d,f). Figure 13 shows the fracture surface of cold-worked specimens with surface-cracks at −180 °C. Mostly brittle fracture and a small proportion of dimples were observed in the 5% and 10% cold-worked specimens without surface-cracks (Figure 13a,c), and 30% cold-worked specimens also showed almost brittle fractures as shown in Figure 13e. However, in case of the cold-worked steels with surface-cracks, the fraction of ductile fracture increased due to the effect of surface-cracks after a surface-cracking technique even at very low temperatures (Figure 13b,d,f). Therefore, the results of the Charpy impact test and fracture surface analysis were in good agreement with the effect of enhanced impact toughness by a surface-cracking technique.

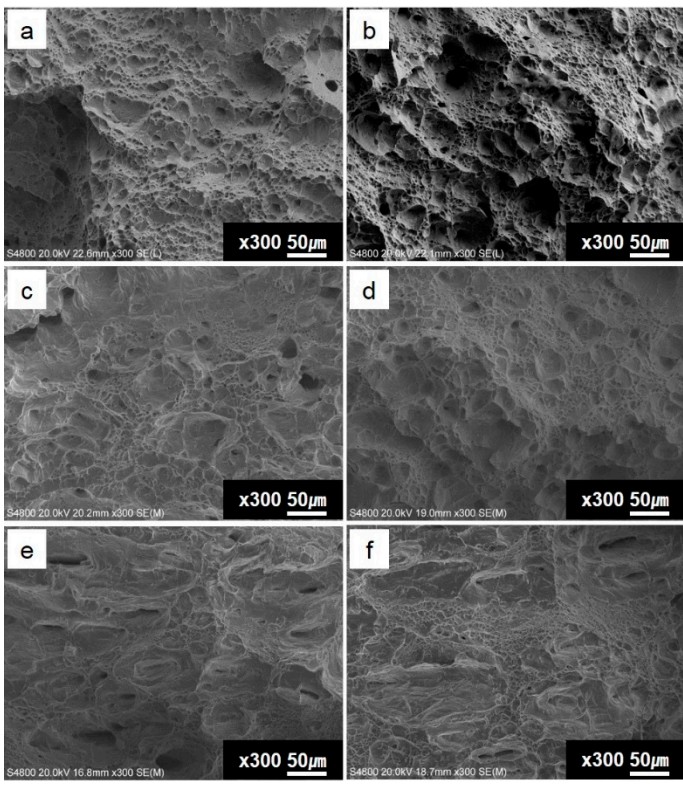

**Figure 12.** Fracture surface of cold-worked specimens without or with surface-cracks at room temperature (20 °C): (**a**) BM-CW05, (**b**) BM-CW05-L10, (**c**) BM-CW10, (**d**) BM-CW10-L10, (**e**) BM-CW30, and (**f**) BM-CW30-L10.

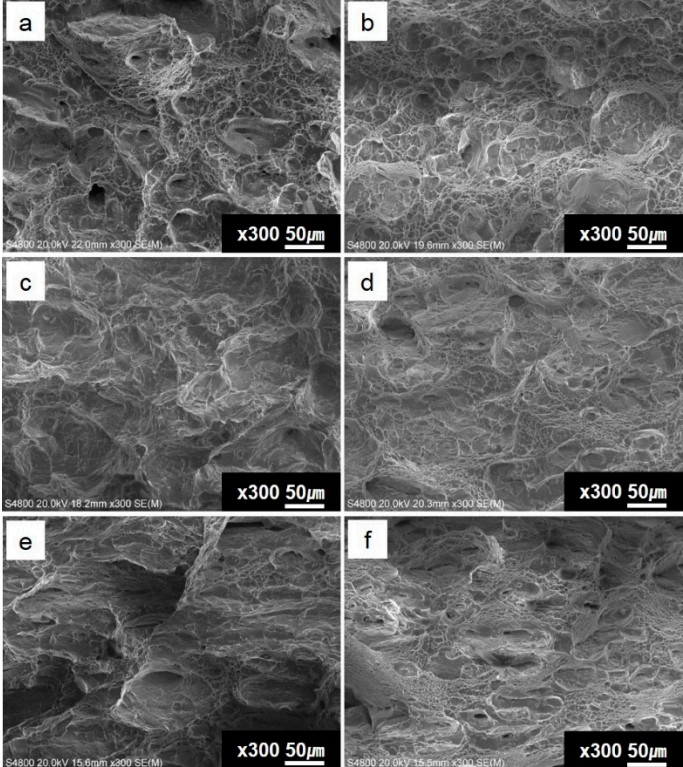

**Figure 13.** Fracture surface of cold-worked specimens without or with surface-cracks at cryogenic temperature (−180 °C): (**a**) BM-CW05, (**b**) BM-CW05-L10, (**c**) BM-CW10, (**d**) BM-CW10-L10, (**e**) BM-CW30, and (**f**) BM-CW30-L10.

*3.6. Application of a Surface-Cracking Technique in Weld Metal*

From the results of the experiments, the surface-cracking technique seems to be effective on some brittle materials rather than the materials with fully ductile properties. In particular, it is very important to improve the mechanical properties of the weld metal because the weld metal is more brittle than the base metal. Therefore, we applied the surface-cracking technique to improve the impact toughness of weld metal. The absorbed energy vs. test temperature curves for welded specimens with surface-cracks is shown in Figure 14. The absorbed energy of welded specimens had a relatively lower value compared to the as-received specimen. After a surface-cracking process, the absorbed energy increased through the effect of surface-cracks at all temperature regions. As the number of surface-cracks increased from 0 lines to 10 lines, the absorbed energy significantly increased. Similar to the results of the cold-worked specimen, the effect of enhanced toughness in welded specimens with surface-cracks happened at all temperature regions. It was clear that this effect occurred in brittle conditions such as low temperatures, cold-worked steel, and weld metal.

This surface-cracking technique has enormous potential for improving the mechanical properties of structure materials. For example, this technique is also applicable to corner areas of rectangular structures. The rectangular structure has a high risk of fracture due to the concentration of stress on the corner areas. It is possible to improve the toughness by inducing surface-cracks at the corner of rectangular structures. We can also control the mechanical properties with this technique because they are dependent on a cold-working level and surface-cracks. The application of this technique to industrial products is designed to change their physical properties and thus improving the products' toughness, resistance, performance, and durability. This may offer many benefits to the optimized design of high-strength ductile metallic materials in the future.

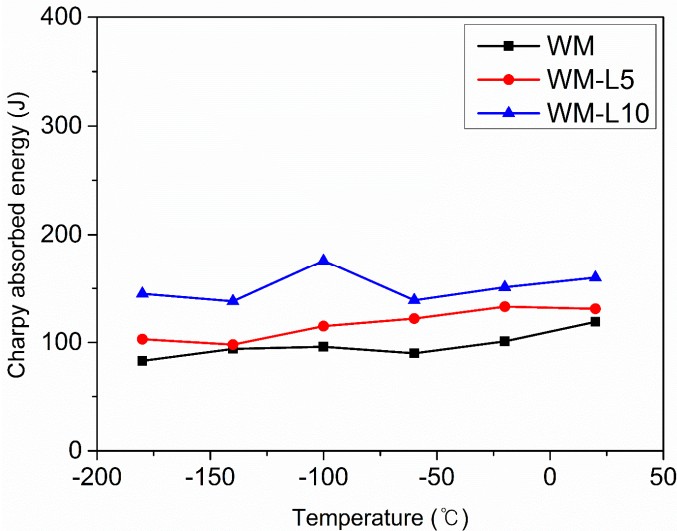

**Figure 14.** The absorbed energy vs. test temperature curves for welded specimens with surface-cracks.

## 4. Conclusions

In this study, we have developed a new technique using a cold-working process and surface-cracking process to increase the strength and toughness of stainless steels at the same time. This surface-cracking technique has an enormous potential for improving the mechanical properties of structure materials. We can also control the mechanical properties with this technique because they are dependent on cold-working levels and surface-cracks. The application of this technology in industrial products is designed to change their mechanical properties and thus improving the products' toughness, strength, and integrity. This may offer many benefits in the optimized design of high-strength ductile metallic materials in the future. The main results obtained in this study are as follows:

1. After a cold-working process, the microstructure was changed from austenite single phase to dual phases which consisted of austenite and martensite. The tensile strength and hardness increased with an increase in the cold-working level. The phase transformation from austenite to martensite and the increase in dislocations by plastic deformation led to the strengthening of cold-worked STS304 steels.

2. The specimen with surface-cracks had a higher absorbed energy compared to the as-received specimen without surface-cracks at low temperature regions (below −60 °C). The dissipation of the crack propagation energy with surface-cracks enhanced the impact toughness, showing a ductile fracture mode even in the cryogenic temperature region.

3. The absorbed energy of cold-worked steels decreased with a cold-working level. Comparing the surface-cracking effect of specimens with the same cold-working level, the absorbed energy of specimens with surface-cracks was two times higher than that of the specimen without surface-cracks. In particular, the strength and toughness were simultaneously increased at 5% cold-worked specimens with surface-cracks. The absorbed energy of the welded specimen had a relatively lower value compared to the as-received specimen. After a surface-cracking process, the absorbed energy was increased by effect of a surface-cracking process. Similar to the result of the cold-worked specimen, toughening by a surface-cracking process in the welded specimen happened in all temperature regions.

**Author Contributions:** K.K. and B.J.K. conceived and designed the experiments; K.K. and M.P. performed the experiments; J.J., H.C.K., H.-S.M., D.-H.L., J.B.J., H.K., S.-H.K., and B.J.K. analyzed and discussed the data; K.K. and B.J.K. wrote the paper.

**Funding:** This study was supported by the R&D Program of the Korea Institute of Industrial Technology (KITECH) as "Fundamental research and development on three-dimensional nano-porous catalysts and gas sensor for hydrogen production and detection" (KITECH EO-18-0022).

**Conflicts of Interest:** The authors declare no conflict of interest.

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
