# Peer review of "Improvement of Strength and Impact Toughness for Cold-Worked Austenitic Stainless Steels Using a Surface-Cracking Technique"

_metals, doi:10.3390/met8110932_

Round 1

Reviewer 1 Report

The paper has been amended and enriched addressing satisfactorily the review comments. The References’ list was also enhanced and experimental details together with fracture analysis results were mainly complemented in the final version of the paper.  

Author Response

Thank you for your useful comments and suggestions on the language and structure of our manuscript. We try to clarify the content and results of our manuscript according to reviewers comments. We modified our manuscript to clarify English language and style.

Reviewer 2 Report

Manuscript metals-386635, "Improvement of strength and impact toughness for cold-worked austenitic stainless steels using a surface-cracking technique"

This paper presents an investigation on the impact toughness of stainless steels STS304 and STS308L depending on the test temperature; kind of surface; and thickness reduction. The work is well done, nevertheless remain some inaccuracies that can be easily clarified, so, and because the paper is well written and do not present critical errors, the manuscript is recommended for publication.

Various comments will be useful for the authors:

1.     Page 3. Lines 103-104. “Charpy V-notch...Figure 1”. Please, rephrase. The syntax is not good.

2.     Page 3. Line 110. “STS304, a plate of STS304…". The language again. The second "STS304" can be "steel". Don't need to repeat.

3.     Page 3. Figure 1. First image (top, left). The symbol of degree is out of position.

In this image, the marking line of the surface cracks appear to be in the opposite side of the V-notch. This happen because, in this drawing, the area near the V is marked with sloping lines. In a technical drawing, these lines must be bellow the line of the surface cracks.

The drawings in the middle of the image are the 2 dimensional technical drawings of the samples. We don't need it because this image also have the 3d drawing. We only need one kind (2d or 3d represents the same).

4.     Page 4. Line 122. “depth of 0.1 mm…”. Where is this dimension at Figure 1? I only see 0.3 mm.

5.     Page 5. Line 156. Please, rephrase. The syntax is not good.

6.     Pages 5-6. Line 157-158. “…the volume fraction of martensite increased about 8% compared to the as-received specimen”. This is not true. The true is: "the volume fraction of martensite increased to about 8%".

7.     Page 6. Lines 158-159. “Therefore…”. Please, rephrase. The syntax is not good.

8.     Page 8. Line 221. “Figure 10e-f” - don't exist.

9.     Page 8. Lines 232-233. Please, rephrase. The syntax is not good.

10.  Page 10. Line 247 “Chapter 3.2” – it is “Section 3.2”.

Author Response

Thank you for your useful comments and suggestions on the language and structure of our manuscript. We try to clarify the content and results of our manuscript according to reviewers comments. We modified our manuscript to clarify English language and style. And detailed corrections are listed below point by point:

1. Page 3. Lines 103-104. “Charpy V-notch...Figure 1”. Please, rephrase. The syntax is not good.

->We have modified the English language in our manuscript.

2. Page 3. Line 110. “STS304, a plate of STS304". The language again. The second "STS304" can be "steel". Don't need to repeat.

->Our manuscript has been modified according to your comments.

3. Page 3. Figure 1. First image (top, left). The symbol of degree is out of position.

In this image, the marking line of the surface cracks appear to be in the opposite side of the V-notch. This happen because, in this drawing, the area near the V is marked with sloping lines. In a technical drawing, these lines must be bellow the line of the surface cracks.

The drawings in the middle of the image are the 2 dimensional technical drawings of the samples. We don't need it because this image also have the 3d drawing. We only need one kind (2d or 3d represents the same).

->Figure 1. has been modified according to your comments.

4. Page 4. Line 122. “depth of 0.1 mm”. Where is this dimension at Figure 1? I only see 0.3 mm.

->Figure 1. has been modified according to your comments.

5. Page 5. Line 156. Please, rephrase. The syntax is not good.

->We have modified the English language in our manuscript.

6. Pages 5-6. Line 157-158. “the volume fraction of martensite increased about 8% compared to the as-received specimen”. This is not true. The true is: "the volume fraction of martensite increased to about 8%".

->Our manuscript has been modified according to your comments.

7. Page 6. Lines 158-159. “Therefore”. Please, rephrase. The syntax is not good.

->We have modified the English language in our manuscript.

8. Page 8. Line 221. “Figure 10e-f” - don't exist.

->Our manuscript has been modified according to your comments.

9. Page 8. Lines 232-233. Please, rephrase. The syntax is not good.

->We have modified the English language in our manuscript.

10. Page 10. Line 247 “Chapter 3.2” it is “Section 3.2”.

->Our manuscript has been modified according to your comments.

Reviewer 3 Report

The present study describes a new technique of combined processes of cold-working and surface cracking in order to improve both the strength and toughness of stainless steels. The manuscript describes a high-level research and provides adequat design, results description and discussion.

Author Response

(The authors gave the same response as above.)
